# A Comprehensive Analysis of Smartphone GNSS Range Errors in Realistic Environments

**DOI:** 10.3390/s23031631

**Published:** 2023-02-02

**Authors:** Jiahuan Hu, Ding Yi, Sunil Bisnath

**Affiliations:** Department of Earth and Space Science and Engineering, York University, Toronto, ON M3J 1P3, Canada

**Keywords:** smartphone range errors, realistic driving scenarios, environment classification, error distribution and correlation

## Abstract

Precise positioning using smartphones has been a topic of interest especially after Google decided to provide raw GNSS measurement through their Android platform. Currently, the greatest limitations in precise positioning with smartphone Global Navigation Satellite System (GNSS) sensors are the quality and availability of satellite-to-smartphone ranging measurements. Many papers have assessed the quality of GNSS pseudorange and carrier-phase measurements in various environments. In addition, there is growing research in the inclusion of a priori information to model signal blockage, multipath, etc. In this contribution, numerical estimation of actual range errors in smartphone GNSS precise positioning in realistic environments is performed using a geodetic receiver as a reference. The range errors are analyzed under various environments and by placing smartphones on car dashboards and roofs. The distribution of range errors and their correlation to prefit residuals is studied in detail. In addition, a comparison of range errors between different constellations is provided, aiming to provide insight into the quantitative understanding of measurement behavior. This information can be used to further improve measurement quality control, and optimize stochastic modeling and position estimation processes.

## 1. Introduction

Over the last decade, the proliferation of low-cost GNSS-enabled smartphones has boosted personalized Location-Based Services (LBS) businesses thanks to the evolution of GNSS systems and the maturity of microelectronic technologies. In 2016, Google released the Android 7.0 platform that allows smartphone users to access the code and phase measurements free of charge [1], enabling researchers to analyze data quality and refine positioning algorithms. This progress has, in turn, significantly expanded smartphone-based mass market applications such as cadastral surveying, asset management, mobile mapping, seismic monitoring, precise agriculture, lane-level navigation for unmanned vehicles, etc. [2,3,4,5].

In addressing demands from multiple industries, how to provide real-time continuous, accurate, and reliable smartphone navigation services has been seen as the major challenge faced by the GNSS community. Owing to hardware limitations, most early studies attempted different strategies to evaluate and improve positioning accuracy of single-frequency smartphones. Pesyna et al. [6] achieved centimetre-level accuracy with smartphone-quality antenna and single-frequency carrier-phase differential GNSS technology. In addition, after phase measurements are available, Sikiria et al. [7] evaluated the single point positioning (SPP) performance with Huawei P10 pseudorange measurements, and obtained horizontal and vertical rms of 10 m and 20 m, respectively. Similar smartphone single point positioning was also seen in [8,9]. However, the positioning accuracy of several metre-level cannot respond to growing market needs. In this context, a host of studies demonstrated that single-frequency smartphone RTK/NRTK solutions may achieve decimetre- to centimetre-level positioning accuracy under ideal environments [10,11,12,13]. Furthermore, Gill et al. [14] carried out single-frequency precise point positioning (PPP) static experiments, and the final solutions provide 37 cm and 51 cm rms in horizontal and vertical domains, respectively. In addition, such solutions can be further improved with the ionospheric-constrained PPP strategy [15].

In 2018, the launch of the first dual-frequency smartphone MI 8 equipped with a BCM47755 chip provided the opportunity to also utilize the L5/E5 frequency and better manage ionospheric delays [16,17,18]. Aggrey et al. [19] compared PPP performance for four smartphones under static and kinematic experiments, and the MI 8 achieved 40 cm rms in the horizontal direction, which was far better than other single-frequency smart devices. In addition, similar performance can be also seen from numerous recent contributions with real-time and final products [20,21,22,23,24]. Continuing this research, recent studies prove that dual-frequency smartphones can provide lane-level navigation processed with both RTK and PPP technologies in realistic driving environments [25,26], and the solutions can be further improved with the aid of smartphone native Inertial Measurement Unites (IMUs) [27,28].

Despite this considerable progress, the major limitations preventing smartphone-based precise applications are their noisy or non-existing measurements due mainly to the poor polarized antenna and multipath contamination [5,6,29]. To tackle these issues, several studies thus far have investigated smartphone signal strength, stochastic modeling, observation noise characterization, as well as measurement optimization strategies. Compared to geodetic GNSS receivers, it is observed that smartphone observations are prone to significantly lower carrier-to-noise density ratios ($C/N_{0}$) [30,31,32], especially for highly multipath-contaminated measurements, indicating that smartphone GNSS modules suffer from poor signal reception. Therefore, it is generally agreed in the literature that the $C/N_{0}$-based weighting scheme is more appropriate for smartphone positioning compared to the elevation-based weighting scheme [33,34,35]. To mitigate the negative impacts of native smartphone antennas, Riley et al. [36] revealed that smart devices with different grades of GNSS antennas are likely to perform significantly better in localization and provided insight into GNSS antenna replacement. Subsequent contributions confirm that smartphone centimeter-level solutions are achievable with an external geodetic antenna through relative positioning [37] and PPP-AR [38].

As unexpected smartphone measurement noise may reach a dozen meters [5,31,35,39], smartphone observation quality assessment has received considerable attention from researchers. Lachapelle et al. [40] evaluated the pseudorange quality from Huawei P10 through code and phase measurements difference, and results illustrate that smartphone code measurements are ten times noisier than geodetic GNSS receivers. Similar conclusions can be drawn from [30,32,41]. Moreover, Liu et al. [31] adopted a short baseline and the single-difference approach between the reference station and smartphone observations to assess the pseudorange residuals including multipath errors and observation noise, and this study showed that there are 10–30 m gross errors existing in smartphone code measurements compared to geodetic receivers, and vary between different constellations. Furthermore, by utilizing the multipath combination algorithm [42], it is confirmed that L5/E5a signals are significantly less influenced by the multipath effect compared to L1/E1 signals [43], owing to their higher transmission power level.

It is well recognized that receiver noise and multipath effects significantly limit the smartphone’s precise positioning and, therefore, its applications. However, in view of all work that has been mentioned, few studies focus on smartphone data quality analysis in realistic, e.g., driving environments, and many experiments are stabilized on open rooftops or vehicle roofs, which do not correspond to consumer habits in daily life. To improve understanding of smartphone measurement behavior and further enhance the stochastic modeling and estimation processes, this study provides a detailed assessment of the actual smartphone range errors under different driving scenarios. Furthermore, no contributions have explored and the relationship between range errors with satellite geometry, signal strength, as well as pre-fit residuals under multiple multipath profiles, and the range errors and their distribution difference caused by the different smartphone mounting location are still unclear so far. Given that, the main contributions of this work are to answer the following questions: (1) How does the smartphone range error distribution behave based on different signal frequencies and constellations? (2) What is the relationship among range errors, satellite elevation angles, and SNR values? (3) How does range error vary under different multipath profiles in real driving environments? Do any differences exist with different mounting location? (4) What is the correlation between smartphone pre-fit residuals and range errors? The novelties of this paper are in analyzing the range errors under different environments, investigating the correlation between pre-fit residuals and range errors, and also comparing the range errors for smartphones mounted on a car roof and dashboard. It is also worth mentioning that, to determine actual range errors, the geodetic tightly-coupled post-processing kinematic (PPK)+inertial measurement unit (IMU) solutions are used as reference, which is different from the smartphone GNSS-based position estimates used in most literature.

This contribution is organized as follows: first, two different range error computation methodologies are comprehensively discussed in Section 2, followed by the measurement campaigns and experimental design in Section 3. Section 4 provided a comprehensive analysis of range errors to address the posed questions. Finally, this paper ends with conclusions and future work.

## 2. Methodology

To calculate the range errors for the smartphone, a collocated geodetic-grade receiver is needed as a reference, for which the range errors can be neglected compared with noisier smartphone measurements. Figure 1 illustrates the receiver sets (*A* and *B*) mounted on the car and with the tracked satellites *m* and *n*. The range errors can then be generated using measurement-differenced or state-differenced methods. In the measurement-differenced method, double-differencing is conducted with raw observations from two close receivers with consideration of the antenna lever arm correction, while in the state-differenced method, the state terms such as slant ionospheric effect, zenith troposphere delay are derived using the un-combined and un-differenced Precise Point Positioning (PPP) model from the reference receiver and applied to the smartphone as “true” values, and then single-differenced state residuals are computed to eliminate receiver-related terms, and hence the range errors are obtained. It should be noted that, due to the differencing procedure, the calculated range errors would be enlarged correspondingly.

The pseudorange measurement *P* on *j*th frequency for satellite *m* at receiver *A* can be given as Equation (Equation 1):(1)$$P_{A,j}^{m}=\rho_{A}^{m}+c\left(dt_{A}-dt^{m}\right)+I_{A,j}^{m}+T_{A}^{m}+b_{A,j}-b_{j}^{m}+\epsilon_{A.j}^{m}$$
where ρ denotes the geometry distance between satellite and receiver, *c* is the speed of light, $dt_{A}$ and $dt^{m}$ are the clock offset for receiver *A* and satellite *m*, respectively. $I_{A,j}^{m}$ is the slant ionospheric effect in meters on *j*th frequency, and *T* is the slant troposphere effect in meters. $b_{A,j}$ is the code hardware delay from the receiver antenna to the signal correlator, and $b_{j}^{m}$ is the code hardware delay from the satellite signal generator to the satellite antenna. $\epsilon_{A.j}^{m}$ is the measurement noise, which contains the range errors that need to be derived.

With another receiver *B*, the double-differenced pseudorange measurement between receiver *A* and *B* and satellite *m* and *n* on *j*th frequency can be given as Equation (Equation 2): (2)$$\nabla ▵P=\left(\rho_{A}^{m}-\rho_{A}^{n}\right)-\left(\rho_{B}^{m}-\rho_{B}^{n}\right)+\delta I+\delta T+\left(\epsilon_{A}^{m}-\epsilon_{A}^{n}\right)-\left(\epsilon_{B}^{m}-\epsilon_{B}^{n}\right)$$
where the ∇ denotes the between-receiver differencing and ▵ denotes the between-satellite differencing, δI and δT are the ionosphere and troposphere residuals after double-differencing, respectively. For two close receivers whose lever arm is within several meters, the atmospheric delays can be eliminated by performing double-differencing, thus δI and δT can be ignored.

With two assumptions, that (a) the range errors for geodetic-grade receiver *A* can be ignored compared to those of smartphone *B* [44], and (b) the satellite with the highest elevation angle which is chosen as reference satellite when performing between-satellite differencing has significant lower range errors compared to lower-elevation satellites, the measurement-differenced range errors can therefore be derived as Equation (Equation 3):(3)$$rangeerror=\nabla ▵P-\left(\rho_{A}^{m}-\rho_{A}^{n}\right)+\left(\rho_{B}^{m}-\rho_{B}^{n}\right)$$
where the geometry distance $\rho^{m}$ can be given as a function of receiver position (x,y,z) for satellite *m*$\left(x^{m},x^{m},z^{m}\right)$ as Equation (Equation 4):(4)$$f\left(x,y,z\right)=\rho^{m}={\left({\left(x-x^{m}\right)}^{2}+{\left(y-y^{m}\right)}^{2}+{\left(z-z^{m}\right)}^{2}\right)}^{1/2}$$

In a realistic environment, the lever arm between *A* and *B* is constant in the vehicle body coordinate system (east, north, and up), and can be transformed to epoch-wised ECEF coordinate system using the approximate position of vehicles. Take one specific epoch for an example, with the coordinates of receiver *A*$\left(X_{A},Y_{A},Z_{A}\right)$, satellite *m*$\left(x^{m},x^{m},z^{m}\right)$ and the lever arm $\left(X_{0},Y_{0},Z_{0}\right)$, Taylor’s formula can be applied to $\rho_{B}^{m}$ at the approximate position of *A* as Equation (Equation 5):(5)$$\rho_{B}^{m}{|}_{\left(x,y,z\right)=\left(X_{A},Y_{A},Z_{A}\right)}=\rho_{A}^{m}+f^{{}^{\prime }}\left(x,y,z\right)\cdot \left(X_{0},Y_{0},Z_{0}\right)+f^{{}^{\prime \prime }}\left(x,y,z\right)\cdot \frac{\left(X_{0},Y_{0},Z_{0}\right)}{2}+\cdots$$

The second order of Equation (Equation 5) can then be derived as $\frac{1}{2}\left(\frac{X_{0}}{\rho_{A}^{m}}+\frac{Y_{0}}{\rho_{A}^{m}}+\frac{Z_{0}}{\rho_{A}^{m}}\right)$ and can be ignored because of the meter-level lever-arm. In this way, only first order is taken into consideration, and hence Equation (Equation 3) can be further reparameterized as Equation (Equation 6):(6)$$\begin{array}{c} rangeerror=\nabla ▵P-(\left(\frac{X_{A}-x^{m}}{\rho_{A}^{m}}X_{0}+\frac{Y_{A}-y^{m}}{\rho_{A}^{m}}Y_{0}+\frac{Z_{A}-z^{m}}{\rho_{A}^{m}}Z_{0}\right)\\ -(\frac{X_{A}-x^{n}}{\rho_{A}^{n}}X_{0}+\frac{Y_{A}-y^{n}}{\rho_{A}^{n}}Y_{0}+\frac{Z_{A}-z^{n}}{\rho_{A}^{n}}Z_{0})) \end{array}$$

It can be noted from Equation (Equation 6) that the coefficients of the lever-arm values are the line-of-sight values from receiver *A* to satellites, and corresponding coefficients can be determined reliably even with a ten-meter-accuracy approximate position. Therefore, the measurement-differenced range errors can be derived with only the pseudorange measurements and the fixed lever arm.

Aside from the measurement-differenced method based on Equation (Equation 1), there is another straightforward approach to estimate range errors through pseudorange residual differencing between satellites. This state-differenced method can better show the estimated state characteristics than the measurement-differenced method. Correspondingly, all receiver-dependent states are eliminated with the relative range errors remaining, and the state-differenced or PPP-based range errors can be expressed as Equation (Equation 7):(7)$$▵P=P_{A,j}^{m}-P_{A,j}^{n}=\left(\rho_{A}^{m}-\rho_{A}^{n}\right)+\left(I_{A,j}^{m}-I_{A,j}^{n}\right)+\left(T_{A}^{m}-T_{A}^{n}\right)+\left(\epsilon_{A}^{m}-\epsilon_{A}^{n}\right)$$

Compared to the aforementioned measurement-differenced range errors, the calculated range errors noise is only amplified once from the differencing. However, it requires accurate satellite-dependent states such as geometric ranges, ionospheric delays, as well as tropospheric delays during computation. Furthermore, other terms such as relativity corrections, Sagnac corrections, solid tide corrections, and satellite antenna corrections [45] need to be taken into consideration. A geodetic receiver, NovAtel OEM7, is utilized in the following processing, and satellite-related states are estimated through PPP processing.

## 3. Measurement Campaigns and Experimental Setup

The data for the analysis are collected in and around York University, Toronto, Canada. York University and adjoining areas cover all the aspects of multiple multipath profiles needed for the experiment. The profiles include highways, parking lots, sub-urban with vegetation, overpasses, and open sky environments, each having a unique multipath characteristic and contributing to better appreciating the range error behaviors. Figure 2 shows the aerial and detailed street view photos of these measurement campaigns, and blue arrows indicate the direction.

As shown in Figure 3, the experimental setup contains two sensor suites. The first contains two geodetic GNSS receivers to provide position trajectory references for tested smartphone. One NovAtel OEM7 receiver is mounted on a rooftop as a reference station within a 5 km baseline, whilst the other receiver is connected with a geodetic antenna and mounted inside the vehicle experimental box (see Figure 3a). The second sensor suite includes Xiaomi MI 8 phone, which is fixed with holders and mounted on the vehicle dashboard to mimic real-life applications (see Figure 3b). In addition, it is feasible to mount MI 8 on the top of the experimental box for in-depth comparison and further analysis. The lever arm between the smartphone and the referenced center has been carefully measured.

Table 1 highlights three datasets used in this study, including the test number, collections date, time duration, as well as phone mounted points. Road tests were carried out on two separate days with identical route and different traffic conditions, and the time durations for each test are 31, 28, and 29 min, respectively. These datasets are used and analyzed in the following section:

## 4. Results

The key to further improving the smartphone positioning performance is to understand the smartphone GNSS measurement characteristics. Therefore, analysis regarding range errors on different frequencies and the relationship between signal-to-noise ratio (SNR) values or elevation angles with range errors is first carried out, followed by the assessment of correlations between range errors and pre-fit residuals, range errors under different environments, and comparison of range errors when the smartphone is mounted on the car dashboard and roof. The smartphone data processing strategies have the assumption that the measurement errors should follow a similar distribution as geodetic receivers; therefore, the zero baseline ZIM2-ZIM3 MGEX stations are processed as a reference to provide a clear view of the characteristics of smartphone range errors compared with geodetic receivers. It is worth mentioning that the first (L1 for GPS and GLONASS, E1 for Galileo, and B1 for BDS) and third (L5 for GPS, and E5a for Galileo) frequencies are used in this analysis for both smartphone and geodetic receiver assessments. Hereafter, P1 and P5 are used to denote the first and third frequencies, respectively. In the result analysis, dataset 1 is used in Section 4.1, Section 4.2, Section 4.3 and Section 4.4, and datasets 2 and 3 are used in Section 4.5.

### 4.1. Distribution of Range Errors

The distribution of range errors on the first and third frequencies is first analyzed. As shown in Figure 4, the upper subplots show the distribution of range errors for the smartphone, while the lower panel gives the results for MGEX stations. It can be observed that, for both smartphone and geodetic receivers, the standard derivation (STD) for the third frequency is smaller than the first frequency, with values of 7.3 m on P1 and 2.1 m on P5 for smartphones, and 0.5 m and 0.2 m for those of geodetic receivers, respectively, which indicates that the observation precision on the third frequency is higher. This difference may be due to the fact that the bandwidth of L5 is larger and the transmitted power of L5 is higher than the first frequency. Therefore, the anti-interference ability and anti-noise performance may also be significantly improved for L5 [31,46,47]. The range errors for the geodetic receivers are obeying a nearly zero-mean distribution, while for smartphones, the mean values are not zero, but rather 0.8 m and 0.6 m for the first and third frequencies, respectively. When further investigating the range errors, the STD of range errors for the geodetic receiver on P1 is 0.5 m, considering that the double-differenced measurements are formed when calculating the range errors, the original code precision should be 2 times smaller than the range errors by applying the error propagating law, which is 0.2–0.3 m. This value is consistent with the code precision used in MGEX data processing as in [48], which demonstrates the validity of the proposed range error derivation method.

To provide a clear view of the characteristics of range errors for different constellations, Figure 5 illustrates the mean values and STDs of each constellation for smartphone and geodetic receivers. For GLONASS and BDS, there are no observations for the test smartphone data, and there are only a few GLONASS observations for ZIM2 and ZIM3. It can be concluded from Figure 5 that, among the four constellations, the GLONASS pseudorange measurements have the largest STDs, with values of 11.8 m and 0.7 m for smartphone and geodetic receivers. Therefore, in the GNSS data processing procedure, GLONASS pseudorange measurements are usually de-weighted by two times, and these results on the first frequency are similar to what has been found in [31]. With the four constellations, except for Galileo, the STDs on P5 are nearly 2 times smaller than on P1, which infers that more weight can be put on the third frequency. For this smartphone dataset, the STDs are more than ten times larger compared to the corresponding constellations for geodetic receivers, and hence, for the analysis for range errors on different frequencies of different constellations, the stochastic models of the smartphone observation noise can be adjusted accordingly in the future research.

### 4.2. Range Errors Correlation with Elevation Angle and SNR

Two common weighting schemes are applied to smartphone and geodetic receiver observations, namely SNR-based and elevation angle-based weighting, respectively. These two weighting schemes indicate that measurement quality is highly related to the SNR and elevation angles. Therefore, in this subsection, the relationship between range errors, elevation angle, as well as SNR values, is assessed. Figure 6 depicts the temporal characteristics of range errors and the corresponding relationship with SNR and elevation angles, and different colors represent different satellites. It can be noted again that the range errors on the first frequency are larger than those on the third frequency from the left panel which shows the time series of the range errors. In addition, there is a significant decreasing trend when the SNR increases for smartphone data both for P1 and P5, and a similar situation can be observed when comparing the relationship between range errors and elevation angles for geodetic receivers. Due to the short observation time for the smartphone data (only 30 min), elevation angles are not fully distributed from 0–90 degrees. However, it can also be concluded that, when the satellite tracked by a smartphone is newly rising or falling, the range errors will increase slightly, and therefore the elevation cut-off angle should be set reasonably. For the test ZIM3 observation, no correlation between range errors and SNRs can be found; this may be the reason that the geodetic receiver is under an optimal observation environment, and the SNR values are at a high level, which less affects the range errors.

To further investigate range error correlations, error bars are plotted in Figure 7 to show the mean values and STDs of absolute range errors at different SNR values and elevation angles. The dots and lines denote the mean values and STDs of absolute range errors, respectively. As shown in Figure 7, it can be clearly observed that there is a significant correlation between smartphone range errors and SNR values, especially on the first frequency. Concurrently, a small correlation on the P1 range errors for geodetic receivers can be observed, which is similar to the results in [44]. The small STD for geodetic receivers at low SNR values (around 25) is due mainly to the limited observations with low SNRs. On the other hand, there is a decreasing STD trend on P1 for the geodetic receiver when the elevation angles are increasing, while the mean values and STDs seem to be stable for P5, which have not been mentioned in the previous research.

In view of the existing literature, the SNR cut-off strategy is widely adopted to screen out satellites that are supposed to contain noisy measurements [49]. However, in this work, it is interesting to identify that sometimes these measurements with lower SNR values are even less noisy and may be beneficial for positioning. Furthermore, range error trends vary with different signal frequencies, indicating that weighting schemes need to be tuned and adjusted when applied to different signals. Meanwhile, the elevation angle-based weighting scheme is usually used for geodetic data processing, which might be well suited to the first frequency. However, there is also a potential weighting scheme that can be proposed in future research, to consider both elevation angle and SNR values or to apply different weighting schemes for different frequencies.

### 4.3. Range Errors under Different Environments

GNSS measurements under environments such as urban canyons, overpasses and vegetation usually suffer from poor signal reception, low gain, large multipath errors, and noises, which result in variation and increases in range errors. To investigate this issue, the range errors under three scenarios are analyzed in this subsection to fathom the impact of the environment on it.

As shown in Figure 8, three typical epochs are selected for detailed analysis: (a) in a parking lot, and can be regarded as an open sky area; (b) in a suburban area, where the car was driving close to tall buildings; (c) in a vegetated area, where the car was close to trees on one side of the road. The sky plots on the right show the corresponding epoch, the second of the day (SoD), and range errors for different satellites are presented by different colors and sized dots (the larger the dots, the larger the range errors). It can be seen from Figure 8 that, owing to nearby buildings and trees limiting sky visibility, GNSS signals were blocked significantly, reducing satellite visibility. As expected, the range errors are at a low level when driving in the parking lot, except for satellite R17, whose elevation angle is under 5°. In contrast, range errors increase significantly when the vehicle was in the suburban area, even for signal azimuths without building blockage. While driving in the vegetation area, it seems that the signal blockage by the tree has limited impacts on the range errors of other tracked satellites.

To better understand the range errors under different environments, the whole trajectory of the smartphone data are classified into three scenarios, namely open sky, vegetation, and sub-urban, Table 2 summarizes the overall statistics of the range errors and the 95th per denotes the range errors of exact the 95th percentile. It can be noted that, in the sub-urban environment, the mean values of range errors and corresponding STDs for P1 and P5 are significantly larger than that in open sky and vegetation, which is expected because of the signal blockage and higher-level multipath in the sub-urban. Under the vegetation environment, the range errors are slightly larger than those in the open sky and are consistent with the analysis above, for which the vegetation does not have a significant impact on range errors in comparison to sub-urban.

### 4.4. Comparison between Range Errors and Pre-Fit Residuals

In smartphone GNSS data processing, quality control is vital and more challenging compared to geodetic data. Pre-fit outlier rejection is one effective quality control method. Therefore, in this subsection, the correlation between ranger errors and pre-fit residuals is assessed. As shown in Figure 9, different colors represent different satellites. It can be concluded that the range errors and pre-fit residuals for P1 have a higher correlation than P5, while the range errors and pre-fit residuals for P5 are at a lower level compared with P1. The reason for a higher correlation in P1 could be that the smartphone antenna is designed for the first frequency, and may not be optimized for other frequencies. To further evaluate the correlation satellite-by-satellite, Figure 10 illustrates the relationship among correlation coefficient, elevation angle, and SNR, where each dot represents one satellite and different colors denote the corresponding SNR values. With a lower elevation angle, the correlation coefficients tend to be smaller. Usually, GLONASS satellites have the highest correlation coefficients despite larger range errors. One possible reason is that, for other constellations, due to the high noise level itself for smartphones, the correlation between range errors and pre-fit residuals can be easily hidden within the noise; while there is a larger range error for GLONASS satellites, so the correlation can be more conspicuous. The results indicate that pre-fit residual rejection/de-weighting is worth further investigation and may provide potential solution improvement in future research, as it reflects the range errors with a high confidence level.

### 4.5. Range Errors Comparison on Car Dashboard and Roof

In a realistic application of smartphone positioning, drivers usually mount their devices on the dashboard for navigation or chatting purposes. However, much of the published research conducted experiments under open-sky environments or mounted smartphones on car roofs. Considering that the car roof may also influence the GNSS observation quality, this subsection assesses how the distribution of range errors will be affected when the smartphone is placed on the car roof compared to the car dashboard. Figure 11 depicts the probability distribution of range errors, in which the blue and green bins denote the range errors on the roof and dashboard, respectively. The STD values of range errors on the car roof are 5.1 m and 5.3 m on the dashboard for P1, while they are 2.2 m and 2.4 m for P5, respectively. A clearer and higher centralization for the range errors on the roof is observed compared to those on the dashboard from Figure 11, which infers that the GNSS observation quality is slightly better when the phone is placed on the car roof, and different code precisions may need to be applied when the smartphones are in different places of the car in the data processing.

## 5. Conclusions and Future Work

This study proposes a differencing method between a geodetic grade receiver and a smartphone with consideration of the lever arm, aiming to derive the actual range errors for smartphones under realistic usage. The methodology of the observation differencing is first introduced, and with several datasets collected, for which the smartphone is mounted on the car dashboard and roof, comprehensive investigation and analysis are carried out.

Firstly, question (1) proposed in Section 1 is addressed by assessing the distribution of the range errors and range error characteristics for the different constellations, and it is found that the range errors of smartphones on the first frequency are significantly larger than those of the third frequency, approximately ten times larger than a geodetic receiver/antenna. In addition, the range errors of GLONASS satellites have the largest STD, with values of 11.8 m and 0.7 m for smartphone and geodetic receivers, respectively. For GPS, Galileo, and BDS, the smartphone STD values of range errors are comparable.

Secondly, to answer question (2), the relationship among range errors, elevation angles, and SNR values is evaluated. Consistent with published research, the smartphone range errors are highly dependent on the SNR values, while the range errors of geodetic receivers are significantly correlated with satellite elevation angles. However, the trend on the third frequency is not as obvious as the first frequency, which is worth further investigation.

Thirdly, a vehicle trajectory is classified into different GNSS environments, namely open sky, sub-urban, and vegetation, to assess smartphone range error behaviors under different scenarios which is proposed in question (3). The results for the dataset analyzed indicate that range errors reach a peak in the suburban environment. While in the vegetation environment, the range errors are slightly larger compared to open-sky environments. Furthermore, for the data processed, the tree blockage seems to have little impact on affecting the measurement quality of other satellites.

Fourthly, a comparison between range errors and pre-fit residuals is carried out to address the question (4). A high correlation is found on the first frequency, while it is not significant for the third frequency. By computing correlation coefficient, it is observed that GLONASS measurements have larger correlation coefficients than others. The reason may be that, for other constellations, the correlation can be easily hidden in the observation noise, and it is similar when the third frequency appears to have a lower correlation. When evaluating the correlation satellite-by-satellite, lower elevation and lower SNR values can also cause a low correlation coefficient.

Finally, a range error comparison is conducted with the smartphone placed in different places. The results show that the STD values of range errors for the first frequency are 5.1 m and 5.3 m on the roof and dashboard, respectively, and are 2.2 m and 2.4 m for the third frequency, indicating that GNSS signals are further suppressed and interfered when the smartphone is mounted on the dashboard.

This paper provides a comprehensive assessment of the actual range errors for smartphones, and the results can further benefit the investigation on optimizing weighting schemes and quality control methods for future work, aiming for a higher level of smartphone positioning and application. The main purpose of this paper is to understand the actual range error distribution and characteristics; hence, some preliminary and commonly assessed performances are carried out first (e.g., relationship between range errors and SNR/elevation angle). However, the trends on different frequencies are slightly different, which can further be a reference of the optimization for quality control. In addition, there are new and interesting findings for the range error characteristics under different environments and the correlations between range error and pre-fit residuals, which can potentially benefit smartphone positioning algorithm development. e.g., different code precision and different weight may be applied to different constellations and different frequencies based on the analysis, and the SNR mask and elevation cut-off angle can be adjusted with different datasets. In addition, regarding different range error behavior under different environments, the weighting scheme considering different environments is worth investigating. Furthermore, the prefit residuals can partly reflect the range errors, different thresholds for prefit outlier rejection may be helpful according to this paper. Finally, different code precisions can also be applied when the smartphone is mounted in different places.

## Figures and Tables

**Figure 1 sensors-23-01631-f001:**
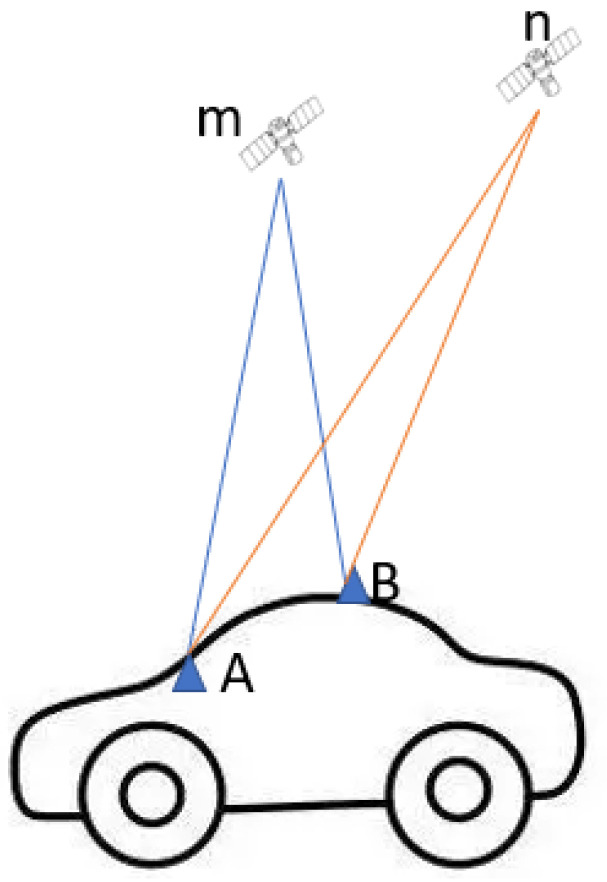
Illustration of receiver suits on a car and the tracked satellites.

**Figure 2 sensors-23-01631-f002:**
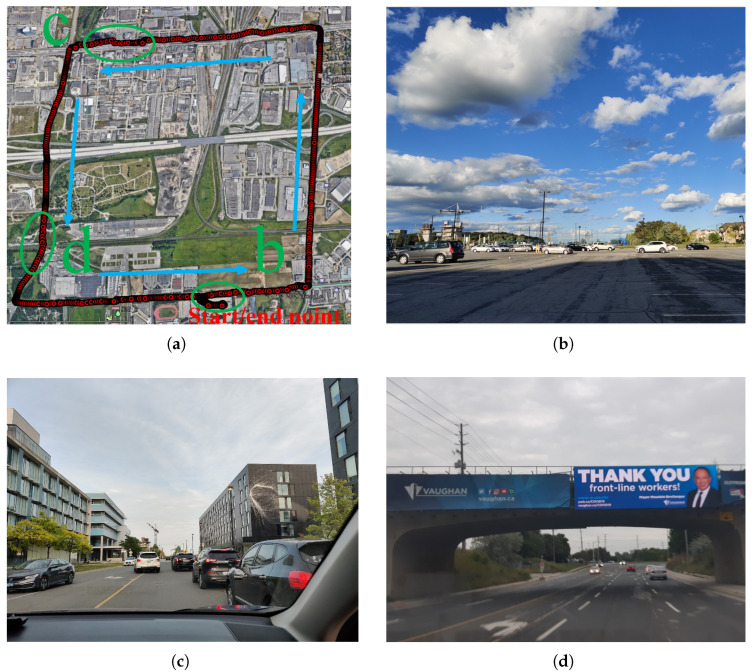
(**a**) aerial and street views of vehicle trajectory with (**b**) open-sky parking lots, (**c**) suburban road, and (**d**) short underpass.

**Figure 3 sensors-23-01631-f003:**
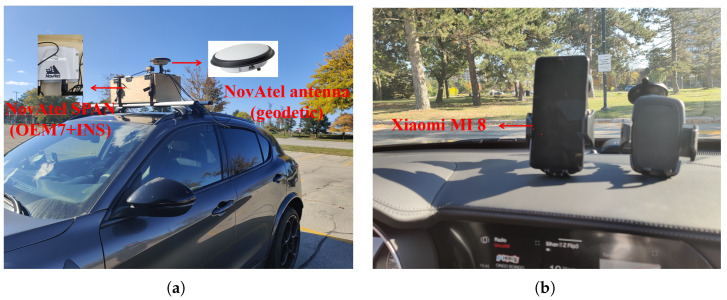
Photos of experimental setup. (**a**) experimental vehicle and setup, (**b**) experimental smartphones.

**Figure 4 sensors-23-01631-f004:**
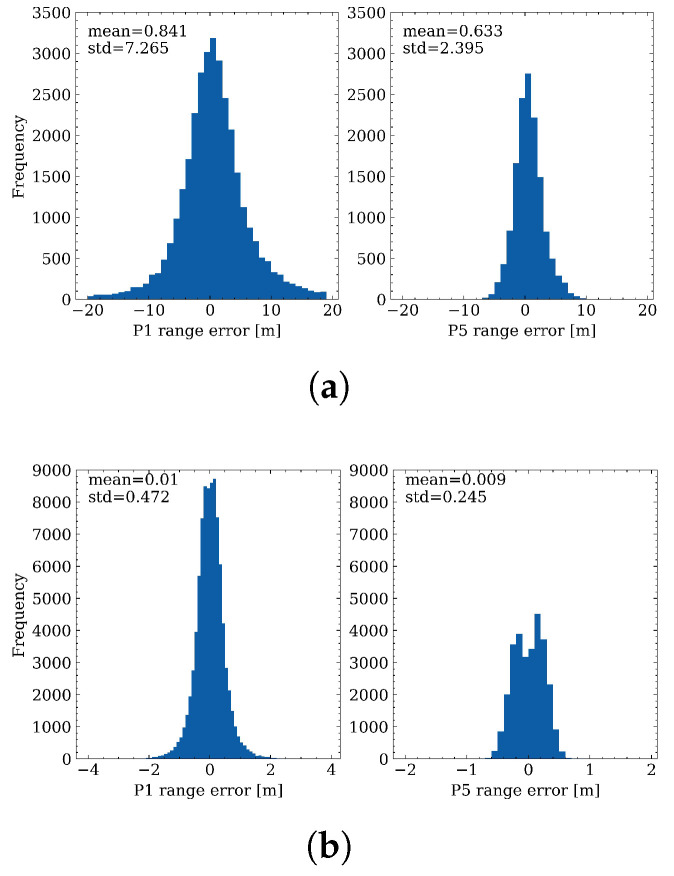
Distribution of range errors on P1 and P5 for different sensors. (**a**) distribution of range errors on smartphones; (**b**) distribution of range errors on MGEX stations.

**Figure 5 sensors-23-01631-f005:**
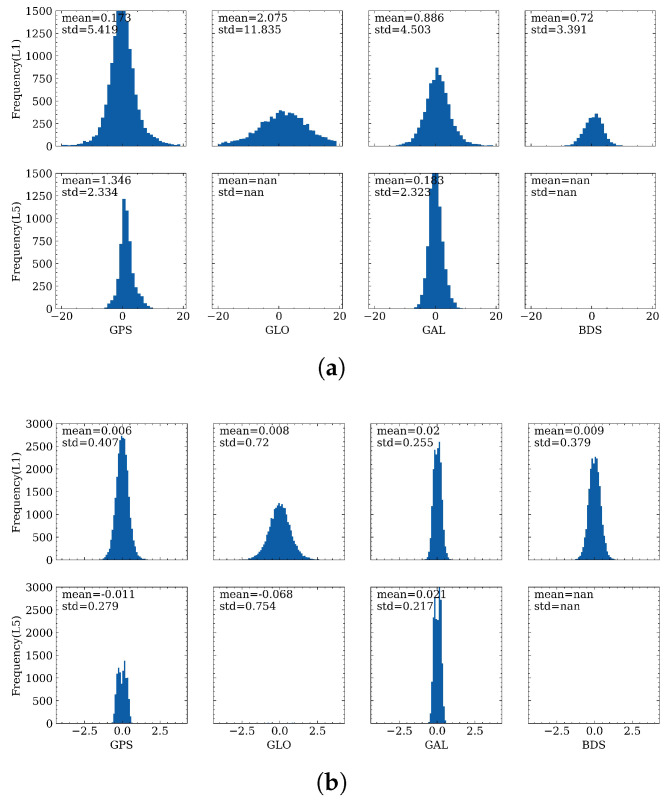
Distribution of range errors of different constellations for (**a**) smartphone and (**b**) ZIM2.

**Figure 6 sensors-23-01631-f006:**
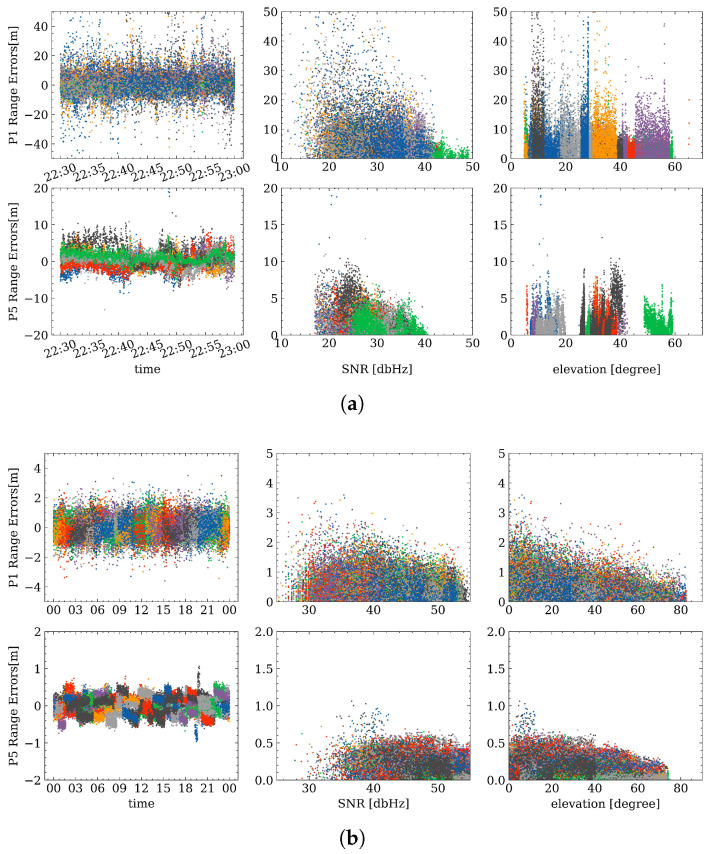
Temporal characteristics of range errors (left) and the relationship with SNR (middle) and elevation angles (right) for smartphone (**a**) and ZIM2 (**b**).

**Figure 7 sensors-23-01631-f007:**
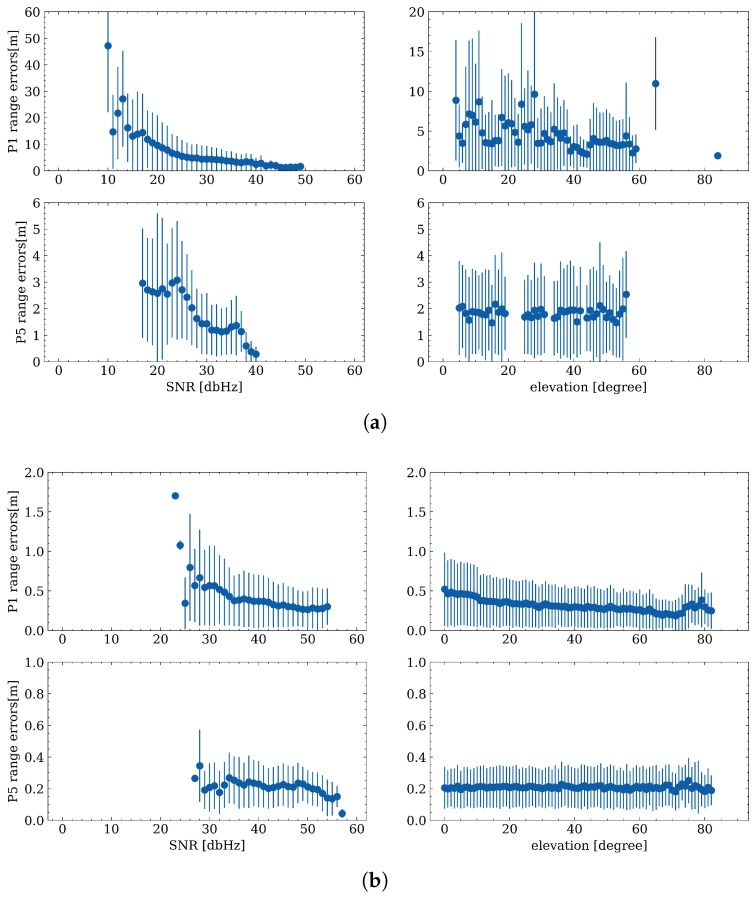
Error bars of range errors changing with SNR (left) and elevation angles (right) for smartphone (**a**) and ZIM2 (**b**).

**Figure 8 sensors-23-01631-f008:**
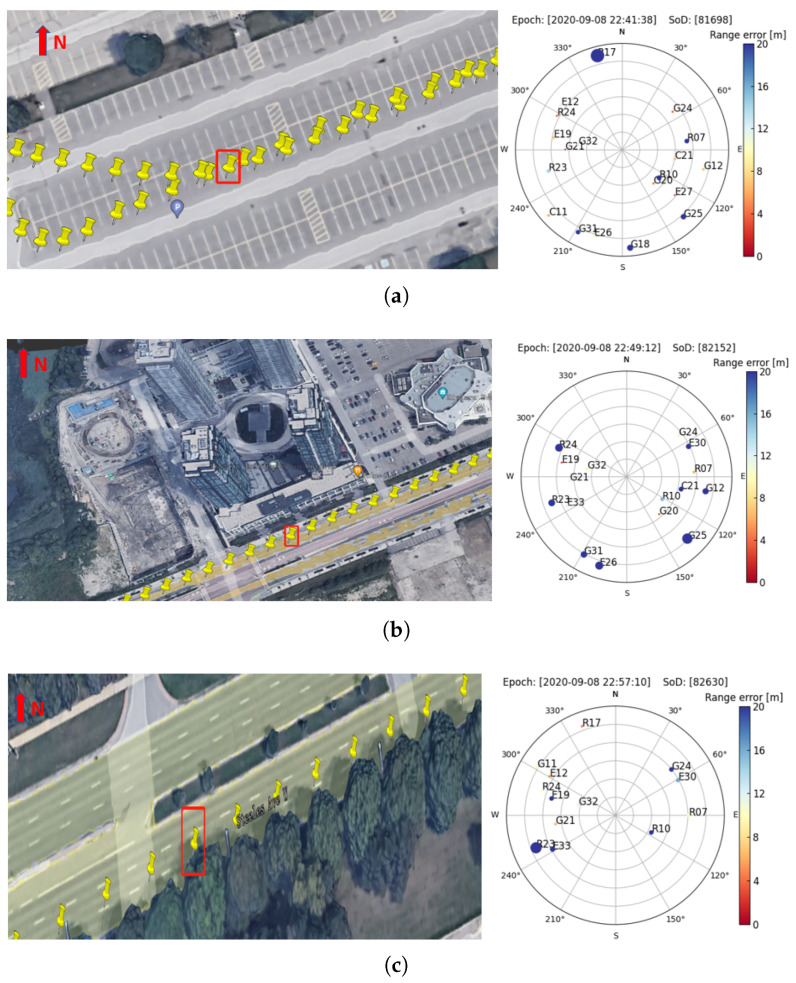
Sky plot of satellite range errors under (**a**) open sky, (**b**) sub-urban, and (**c**) vegetated environments.

**Figure 9 sensors-23-01631-f009:**
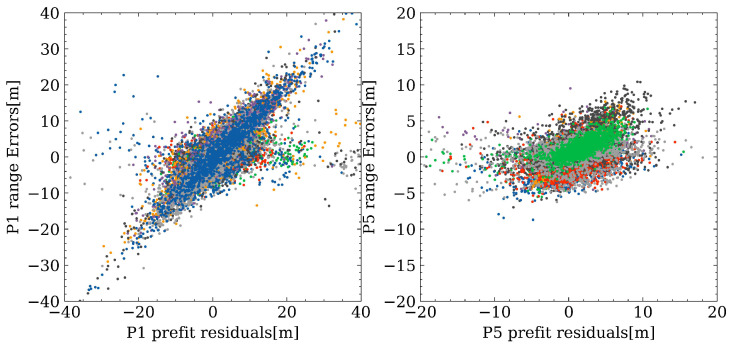
Correlation between range errors and pre-fit residuals.

**Figure 10 sensors-23-01631-f010:**
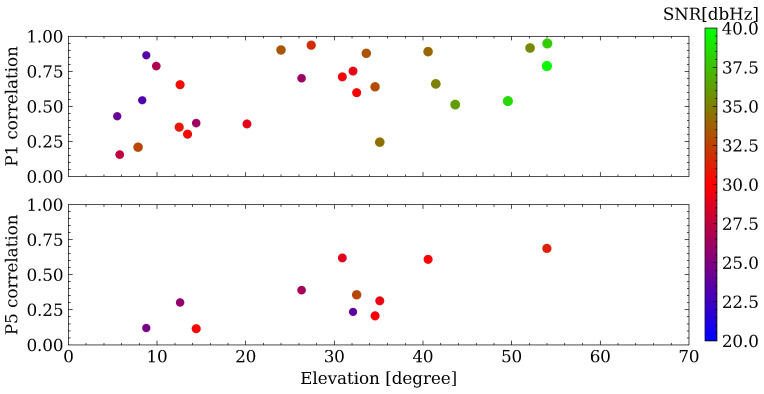
Relationship among correlation coefficients, elevation angle, and SNR values.

**Figure 11 sensors-23-01631-f011:**
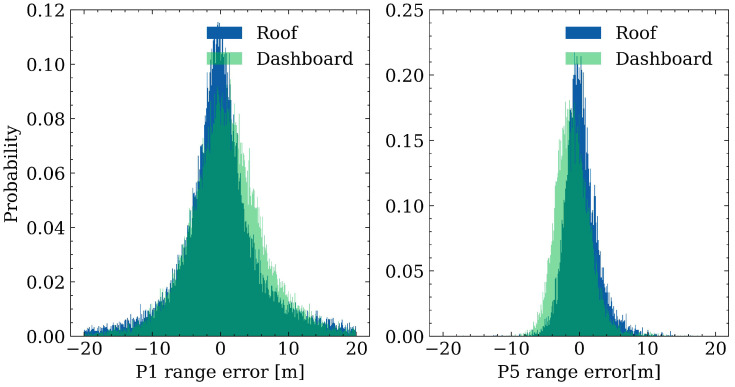
Range error distribution comparison when the smartphone is placed on the dashboard and roof.

**Table 1 sensors-23-01631-t001:** Summary of collected smartphone datasets.

Test #	Collection Date	Duration	Mounted Points
1	8 September 2020	22:29–23:00	Roof
2	8 August 2021	1:14–1:42	Roof
3	8 August 2021	1:50–2.19	Dashboard

**Table 2 sensors-23-01631-t002:** Statistics for satellite absolute range errors under different environments (unit: meter).

	P1				P5			
	Mean	STD	95th per	Number	Mean	STD	95th per	Number
Open-sky	0.8	7.3	14.5	32267	0.9	2.4	5.1	14198
Sub-urban	2.2	9.4	18.2	1443	1.2	3.8	5.9	658
Vegetation	1.5	7.5	17.1	354	1.4	1.9	3.9	176

## Data Availability

The data showed in this study are available from the corresponding author on reasonable request.

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
