# Peer review of "A Comprehensive Analysis of Smartphone GNSS Range Errors in Realistic Environments"

_sensors, 2023, doi:10.3390/s23031631_

Round 1

Reviewer 1 Report

The paper by Hu et al. focuses on evaluating the smartphone range errors in terms of error distribution and characteristics, range errors under different environments, and the relationship between range errors and pre-fit residuals. Overall, this paper contains some investigation which has not been found in other publications (e.g., correlations between range errors and pre-fit residuals), and is of interest to the smartphone positioning community. However, this paper can still be improved, and the comments are as follows.

(1)   In line 111, “the tracked satellites m and n”, please set the m and n in italic type.

(2)   For the paragraph before Equation (2), “Using another receiver B”, please use a more exact word “With another receiver”.

(3)   From line 133, the authors state “with two assumptions….(a)…(b)”, please add proper citations for the two assumptions to support the assumptions are reasonable.

(4)   For paragraph after equation (4), it is stated “the lever arm between A and B is constant in a local coordinate system (east, north, and up)”, in my opinion, it is constant in the body system, and it will change in a local coordinate system when the attitude of the vehicle changes, please check.

(5)   In line 145, please add proper citations for these models.

(6)   In line 148, the authors have mentioned before that the satellite with the highest elevation angle is selected as the reference, therefore, it does not to be replicated here.

(7)   In line 168, “time duration. As well as phone mounted points”, please change the full stop into comma, because the sentence does not end.

(8)   The authors should specify which dataset is used in each sub-section.

(9)   In table 2, what does the “95th per” mean, please explain it in the text.

Author Response

  • In line 111, “the tracked satellites m and n”, please set the m and n in italic type.

A: Thank you for the comments, we have corrected them.

  • For the paragraph before Equation (2), “Using another receiver B”, please use a more exact word “With another receiver”.

A: We have changed it into “with another receiver”

  • From line 133, the authors state “with two assumptions….(a)…(b)”, please add proper citations for the two assumptions to support the assumptions are reasonable.

A: We have added the corresponding citation.

  • For paragraph after equation (4), it is stated “the lever arm between A and B is constant in a local coordinate system (east, north, and up)”, in my opinion, it is constant in the body system, and it will change in a local coordinate system when the attitude of the vehicle changes, please check.

A: Thank you for the comment, we have checked, and it should be in the vehicle body coordinate system, and we also changed it in the manuscript.

  • In line 145, please add proper citations for these models.

A: We have added the reference.

  • In line 148, the authors have mentioned before that the satellite with the highest elevation angle is selected as the reference, therefore, it does not to be replicated here.

A: Thanks, we have removed this sentence.

  • In line 168, “time duration. As well as phone mounted points”, please change the full stop into comma, because the sentence does not end.

A: Thanks, we have changed it.

(8)   The authors should specify which dataset is used in each sub-section.

 A: The use of the dataset is added in the results section as “In the result analysis, dataset 1 is used in 4.1-4.4, and dataset 2 and 3 are used in 4.5.

(9)   In table 2, what does the “95th per” mean, please explain it in the text.

A: We have explained in the text as “and the 95th per denotes the range errors of exact the 95th percentile

Reviewer 2 Report

Dear Authors,

thank you for the interesting Paper on Smartphone Observation Quality.

I do recommend your paper for publication but have two minor remarks that you might consider to improve/clarify before pubplication:

- you mention that the L5 range errors are smaller than the L1 errors because the transmitted power is higher. I think this is wrong. The main reason for the lower noise is that the bandwidth is larger at L5 and therefore the correlator has a steeper discrimination function.

-Figure 5 a) and b) should each get labels "L1" for the first row and "L5" for the second row

Author Response

- you mention that the L5 range errors are smaller than the L1 errors because the transmitted power is higher. I think this is wrong. The main reason for the lower noise is that the bandwidth is larger at L5 and therefore the correlator has a steeper discrimination function.

A: Thank you for the comment. The reason that you give is reasonable, and after reviewing some papers and online sources, we found that both the transmitted power and the larger bandwidth are plausible reasons for L5 being “better” than L1. We have updated the manuscript (as well as the references) to “This difference may be due to the fact that the bandwidth of L5 is larger and the transmitted power of L5 is higher than the first frequency. Therefore, the anti-interference ability and anti-noise performance may also be significantly improved for L5.

-Figure 5 a) and b) should each get labels "L1" for the first row and "L5" for the second row

A: Thank you, we have added the labels and updated the Figure 5 a) and b).

Reviewer 3 Report

1-There are many studies in this field. There is no significant difference between this study and other studies.

2- Most of the results obtained in this study are available in other studies in the literature.

3-Approximately half an hour of measurement duration is not sufficient for this statistical analysis. Satellite geometry does not change significantly in half an hour.

Author Response

1-There are many studies in this field. There is no significant difference between this study and other studies.

2- Most of the results obtained in this study are available in other studies in the literature.

A: Thank you for your comments. We view Comments 1 and 2 as overlapping, so we have combined them and are providing one response. Sorry for causing misunderstanding by not more explicitly describing the novelty in the manuscript. In the Introduction, we have specified “The novelties of this paper are in analyzing the range errors under different environments, investigating the correlation between pre-fit residuals and range errors, and also, comparing the range errors for smartphones mounted on a car roof and dashboard. It is also worth mentioning that to determine actual range errors, the geodetic tightly-coupled post-processing kinematic (PPK)+inertial measurement unit (IMU) solutions are used as reference, which is different from the smartphone GNSS-based position estimates used in most literature.”.

The main purpose for this paper is to understand how the range error distribution, and thus, some similar analysis needs to be carried out first (e.g., relationships between range errors and SNR/elevation angle), and therefore, we also add some references in the text for these parts. However, for all other analyses, it’s the Author’s humble view that they have not been assessed in other publications. Also, there are also new and interesting findings, such as range errors behave differently for different frequencies, and these outcomes can actually be a reference in a further investigation of smartphone GNSS measurement quality control. Therefore, we add a paragraph in the Conclusion to further stress these new findings and the potential benefit. “The main purpose of this paper is to understand the actual range error distribution and characteristics, hence, some preliminary and commonly assessed performances are carried out first (e.g., relationship between range errors and SNR/elevation angle), However, the trends on different frequencies are slightly different, which can further be a reference of the optimization for quality control. Besides, there are new and interesting findings for the range error characteristics under different environments and the correlations between range error and pre-fit residuals, which can potentially benefit smartphone positioning algorithm development.

3-Approximately half an hour of measurement duration is not sufficient for this statistical analysis. Satellite geometry does not change significantly in half an hour.

A: Thank you for the comment. We agree that the satellite geometry would not change much within half an hour. In the data processing, we processed three datasets as shown in our manuscript, therefore, there are 3 x 30 minutes with different data collection geometries. To make the text clear to the reader, we only include one dataset for each subsection.

For this analysis, we are focused not on the transmitter motion (e.g., for carrier-phase ambiguity resolution), but rather the receiver motion with respect to potential near-field reflectors. Even with the relatively short duration data of 30 minutes, the range errors for one satellite can even vary largely with the environment change as shown in Figure 8 (for, e.g., satellites R23, G25, etc.) and this variation is exactly the sampling required to for the manuscript narrative: under different environments, different code precision may be applied.

Round 2

Reviewer 3 Report

The novelties stated in the revised paper are not sufficient for the publication.